# MOSTWAS: Multi-Omic Strategies for Transcriptome-Wide Association Studies

**Arjun Bhattacharya**[1], **Yun Li**[2,3,4], **Michael I. Love**[2,3]*

**1** Department of Pathology and Laboratory Medicine, University of California-Los Angeles, Los Angeles, California, United States of America, **2** Department of Biostatistics, University of North Carolina at Chapel Hill, Chapel Hill, North Carolina, United States of America, **3** Department of Genetics, University of North Carolina at Chapel Hill, Chapel Hill, North Carolina, United States of America, **4** Department of Computer Science, University of North Carolina at Chapel Hill, Chapel Hill, North Carolina, United States of America

* milove@email.unc.edu

**Data Availability Statement:** All relevant data are referred to within the manuscript and its Supporting Information files: MOSTWAS software, https://github.com/bhattacharya-a-bt/MOSTWAS Sample code: https://github.com/bhattacharya-a-

## Abstract

Traditional predictive models for transcriptome-wide association studies (TWAS) consider only single nucleotide polymorphisms (SNPs) local to genes of interest and perform parameter shrinkage with a regularization process. These approaches ignore the effect of distal-SNPs or other molecular effects underlying the SNP-gene association. Here, we outline multi-omics strategies for transcriptome imputation from germline genetics to allow more powerful testing of gene-trait associations by prioritizing distal-SNPs to the gene of interest. In one extension, we identify mediating biomarkers (CpG sites, microRNAs, and transcription factors) highly associated with gene expression and train predictive models for these mediators using their local SNPs. Imputed values for mediators are then incorporated into the final predictive model of gene expression, along with local SNPs. In the second extension, we assess distal-eQTLs (SNPs associated with genes not in a local window around it) for their mediation effect through mediating biomarkers local to these distal-eSNPs. Distal-eSNPs with large indirect mediation effects are then included in the transcriptomic prediction model with the local SNPs around the gene of interest. Using simulations and real data from ROS/MAP brain tissue and TCGA breast tumors, we show considerable gains of percent variance explained (1–2% additive increase) of gene expression and TWAS power to detect gene-trait associations. This integrative approach to transcriptome-wide imputation and association studies aids in identifying the complex interactions underlying genetic regulation within a tissue and important risk genes for various traits and disorders.

## Author summary

Transcriptome-wide association studies (TWAS) are a powerful strategy to study gene-trait associations by integrating genome-wide association studies (GWAS) with gene expression datasets. TWAS increases study power and interpretability by mapping genetic variants to genes. However, traditional TWAS consider only variants that are close to a gene and thus ignores important variants far away from the gene that may be involved in complex regulatory mechanisms. Here, we present MOSTWAS (Multi-Omic Strategies

bt/mostwas_suppdata Models and full results, https://doi.org/10.5281/zenodo.4314067 [32] TCGA GDC Legacy Archive, https://portal.gdc. cancer.gov/legacy-archive GDAC Firehose Browser, https://gdac.broadinstitute.org ROS/MAP data, https://www.synapse.org/#!Synapse:syn3219045 iCOGS GWAS Summary Statistics, http://bcac. ccge.medschl.cam.ac.uk/bcacdata/icogs-complete-summary-results IGAP Late-onset Alzheimer's Disease Risk GWAS Summary Statistics, http:// web.pasteur-lille.fr/en/recherche/u744/igap/igap_ download.php PGC Major Depressive Disorder GWAS Summary Statistics, https://www.med.unc. edu/pgc/download-results/mdd/ UKBB Major Depressive Disorder GWAX Summary Statistics, http://gwas-browser.nygenome.org/downloads/ gwas-browser/ PsychENCODE Project, http:// resource.psychencode.org/.

Funding: A.B. is supported by the National Institute of Environmental Health Sciences (P30-ES010126, https://www.niehs.nih.gov/), Y.L. is partially supported by the National Heart, Lung, and Blood Institute (R01-HL129132, https://www.nhlbi.nih. gov/), the National Institute of General Medical Sciences (R01-GM105785, https://www.nigms.nih. gov/), the National Heart, Lung, and Blood Institute (R01-HL146500, https://www.nhlbi.nih.gov/), and the National Institute of Child Health and Human Development (U54-HD079124, https://www.nichd. nih.gov/). M.I.L. is supported by the National Cancer Institute (P01-CA142538, https://www. cancer.gov/), the National Institute of Environmental Health Sciences (P30-ES010126, https://www.niehs.nih.gov/), and the National Institute of Mental Health (R01-MH118349, https:// www.nimh.nih.gov). The funders had no role in study design, data collection and analysis, decision to publish, or preparation of the manuscript.

Competing interests: The authors have declared that no competing interests exist.

for TWAS), a suite of tools that extends the TWAS framework to include these distal variants. MOSTWAS leverages multi-omic data of regulatory biomarkers (transcription factors, microRNAs, epigenetics) and borrows from techniques in mediation analysis to prioritize distal variants that are around these regulatory biomarkers. Using simulations and real public data from brain tissue and breast tumors, we show that MOSTWAS improves upon traditional TWAS in both predictive performance and power to detect gene-trait associations. MOSTWAS also aids in identifying possible mechanisms for gene regulation using a novel added-last test that assesses the added information gained from the distal variants beyond the local association. In conclusion, our method aids in detecting important risk genes for traits and disorders and the possible complex interactions underlying genetic regulation within a tissue.

## Introduction

Genomic methods that borrow information from multiple data sources, or "omic" assays, offer advantages in interpretability, statistical efficiency, and opportunities to understand causal molecular pathways in disease regulation [1,2]. Transcriptome-wide associations studies (TWAS) aggregate genetic information into functionally relevant testing units that map to genes and their expression in a trait-relevant tissue. This gene-based approach combines the effects of many regulatory variants into a single testing unit that can increase study power and aid in interpretability of trait-associated genomic loci [3,4]. However, traditional TWAS methods, like PrediXcan [3] and FUSION [4], focus on local genetic regulation of transcription. These methods ignore significant portions of heritable expression that can be attributed to distal genetic variants that may indicate complex mechanisms contributing to gene regulation.

Recent work in transcriptional regulation has estimated that distal genetic traits account for up to 70% of gene expression heritability [5,6]. These results accord with Boyle *et al*'s omnigenic model, proposing that regulatory networks are so interconnected that a majority of genetic variants in the genome, local or distal, have indirect effects on the expression level of any particular gene [6,7]. In fact, work by Sinnott-Armstrong *et al* showed huge enrichment of significant genetic signal near genes involved in the relevant pathways for biologically simple traits, even for phenotypes largely thought to be simpler than complex diseases [8]. Together, these observations suggest that the majority of phenotype heritability, even for traits commonly believed to be simpler than complex diseases like cancer, is not driven by variants in core genes, but rather from thousands of variants spreading across most of the genome.

Many groups have leveraged the omnigenic model to identify distal expression quantitative loci (eQTLs) by testing the effect of a distal-eSNP on a gene mediated through a set of genes local to the distal-eSNP, where the SNP and gene are more than 1 Megabase (Mb) away. These studies draw the conclusion that many distal-eQTLs are often eQTLs for one or more of their local genes [9–15]. It has been shown previously that distal-eQTLs found in regulatory hotspots are often cell-type specific [9,13,16] and hence carry biologically relevant signal when studying bulk tissue with heterogeneous cell-types (e.g. cancerous tumors or the brain). More recently, the concepts of distal-eQTLs residing in or near regulatory elements have been integrated with multi-omics data and biological priors to reconstruct molecular networks and hypothesize cell-regulatory mechanisms [17].

Variant-mapping methods have also shown the utility of integrating molecular data beyond transcriptomics. Deep learning methods have been employed to link GWAS-identified variants to nearby regulatory mechanisms to generate functional hypotheses for SNP-trait

associations [18–20]. These ideas have been extended to TWAS: the EpiXcan method demonstrates that incorporating epigenetic information into transcriptomic prediction models generally improves predictive performance and power in detecting gene-trait associations in local-only TWAS [21]. Wheeler *et al* have leveraged TWAS imputation to show that *trans*-acting genes are often found in transcriptional regulation pathways and are likely to be associated with complex traits [22]. Thus, it is imperative to prioritize distal variants that are *trans*-acting to fully capture heritable gene expression that is associated with complex diseases in TWAS.

To this end, we developed two extensions to TWAS, borrowing information from other omics assays to enrich or prioritize mediator relationships of eQTLs in expression models. Using simulations and data from Religious Orders Study and the Rush Memory and Aging Project (ROS/MAP) [23] and The Cancer Genome Atlas (TCGA) [24], we show considerable improvements in transcriptomic prediction and power to detect gene-trait associations. These **M**ulti-**O**mic **S**trategies for **T**ranscriptome-**W**ide **A**ssociation **S**tudies are curated in the R package MOSTWAS, available freely at https://bhattacharya-a-bt.github.io/MOSTWAS.

## Results

### Overview of MOSTWAS

MOSTWAS incorporates two methods to include distal-eQTLs in transcriptomic prediction: mediator-enriched TWAS (MeTWAS) and distal-eQTL prioritization via mediation analysis (DePMA). Here, we refer to an eQTL as a SNP with an association with the expression of a gene, and a distal-eQTL is more than 1 Mb away from the eGene. As large proportions of total heritable gene expression are explained by distal-eQTLs local to regulatory hotspots [6,11,13,14], we used data-driven approaches to either identify mediating regulatory biomarkers (MeTWAS) or distal-eQTLs mediated by local biomarkers (DePMA) to increase predictive power for gene expression and power to detect gene-trait associations. These methods are described in **Methods** with an algorithmic summary in **Fig 1A and 1B** and **S1 Text**.

**Fig 1C** provides an example of the biological mechanisms MOSTWAS attempts to leverage in its predictive models for a gene *G* of interest: here, without loss of generality of the regulatory mechanism, assume a SNP within a regulatory element affects the transcription of gene *X* that codes for a transcription factor. Transcription factor X then binds to a distal regulatory region and affects the transcription of gene *G*. Methodologically,

- MeTWAS first detects the association between the expression of gene *X* and expression of gene *G*. It proceeds upstream in the regulatory pathway to the genetic locus around gene *X* and builds a predictive model for the expression of gene *X* using only SNPs in a local window around it. Imputed expression of gene *X* (imputed via cross-validation) is then included as a fixed effect in the predictive model of gene *G*, along with the genetic variants local to gene *G*. This model is fit using a two-stage regression model [25]: first fitting the imputed mediators using least squares regression and then fitting the local genotypes using elastic net regression [26] or linear mixed modeling [27]. Full details are provided in **Methods** and **S1 Text**.

- DePMA first detects the distal-eQTL association between the distal SNP and expression of gene *G*. It then proceeds downstream in the regulatory pathway from the distal SNP to identify whether there is a strong association between the SNP and the expression of the local gene *X*. Using mediation analysis, DePMA tests if the indirect effect of the SNP on gene *G* mediated through gene *X* is significantly large. If so, the SNP is included in the final predictive model for the expression of gene *G*. All local SNPs to gene *G* and significantly mediated distal-eQTLs are used to fit a predictive model for gene *G* using elastic net regression [26] or linear mixed modeling [27]. Full details are provided in **Methods** and **S1 Text**.

**A. MeTWAS scheme**

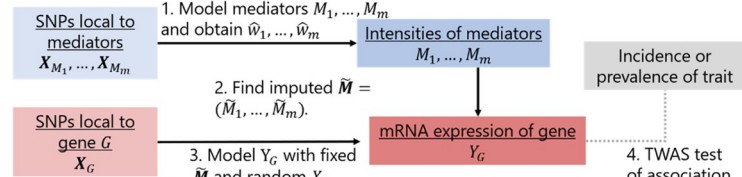

**B. DePMA scheme**

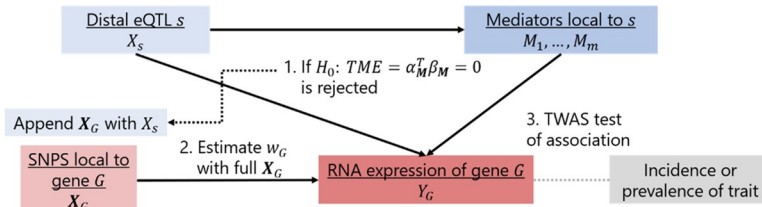

**C. Example biological mechanism leveraged by MOSTWAS**

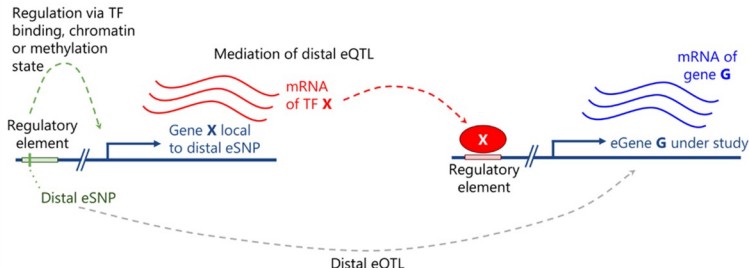

**Fig 1. *Overview of MOSTWAS methods and biological context*. (A)** Overview of Mediator-enriched TWAS (MeTWAS). We build local predictive models of distal mediators associated with the gene of interest (Step 1). Next, we imputed mediator intensities into the training step (Step 2) and fit the final predictive model with imputed mediator intensities as fixed effects and local SNPs to the gene as regularized effects (Step 3). This model can be used for TWAS (Step 4). **(B)** Overview of Distal-eQTL Prioritization via Mediation Analysis (DePMA). We test distal-eQTLs for their indirect mediation effect on the gene through mediators local to the distal-eSNP (Step 1). We append the set of local SNPs to the gene with highly mediated distal-eSNPs and fit the final model with regularized regression (Step 2). This model can be used for TWAS (Step 3). **(C)** Example of a biological mechanism MOSTWAS leverages in its predictive models. Here, assume a SNP (in green) within a regulatory element affects the transcription of gene *X* that codes for a transcription factor. Transcription factor *X* then binds to a distal regulatory region and affects the transcription of gene *G*. The association between the expression of gene *X* and gene *G* is leveraged in the first step of MeTWAS. A distal-eQTL association is also conferred between this distal-SNP and the eGene *G*, which is leveraged in the DePMA training process.

MeTWAS and DePMA can consider any set of regulatory elements as potential mediators (e.g. transcription factors, microRNAs, CpG methylation sites, chromatin-binding factors, etc). For both methods, individual-level genotype data and omic data for mediators and genes of interest are required. Further, MeTWAS requires a list of mediators associated with the gene of interest. This list can be generated from correlation analysis between mediators and genes or through *a priori* knowledge about a particular tissue. DePMA additionally requires distal-eQTL summary statistics and local-xQTL summary statistics between SNPs and mediators.

In MOSTWAS, if expression of a gene has significantly positive germline heritability [28] and the model has five-fold cross validation adjusted $R^2 \geq 0.01$ in predicting observed expression of the gene, then we call the gene model *significant* and it can be used in TWAS. If individual genotype data is available in an external GWAS panel, a MeTWAS or DePMA model

may be used to impute tissue-specific expression. If only summary statistics are available in the GWAS panel, the Imp-G weighted burden testing framework [29] as implemented in FUSION [4] can be applied. We further implement a permutation test to assess whether the overall gene-trait association is significant, conditional on the GWAS effect sizes [4] and a novel distal-SNPs added-last test that assesses the added information from distal-SNPs given the association from the local SNPs (**Methods**).

## Simulation analysis

We first conducted simulations to assess the power to predict gene expression and power to detect gene-trait associations under various settings for phenotype heritability, local heritability of expression, distal heritability of expression, and proportion of causal local and distal SNPs for MeTWAS and DePMA (full simulation details in **Methods**). Using genetic data from TCGA-BRCA as a reference, we used SNPs local to the gene *ESR1* (Chromosome 6) to generate local eQTLs and SNPs local to *FOXA1* (Chromosome 14) to generate distal-eQTLs for a 400-sample eQTL reference panel and 1,500-sample GWAS imputation panel. Though the choice of these loci was arbitrary for constructing the simulation, there is evidence that *ESR1* and *FOXA1* are highly co-expressed in breast tumors, and local-eQTLs of *FOXA1* have been shown to be distal-eQTLs of *ESR1* [30]. We considered two scenarios for each set of simulation parameters: (1) an ideal case where the leveraged association between the distal-SNP and gene of interest exists in both the reference and imputation panel, and (2) a "null" case where the leveraged association between the distal-SNP and the gene of interest exists in the reference panel but does not contribute to phenotype heritability in the imputation panel. We ran 1,000 simulations for every unique set of simulation parameters in both simulation scenarios and computed TWAS power for models trained using local-only modeling (traditional FUSION models), Bayesian Genome-Wide TWAS (BGW-TWAS), a concurrent TWAS method that includes distal variants [31], and MOSTWAS.

In these simulation studies, we found that MOSTWAS methods performed well in prediction across different causal proportions and local and distal mRNA expression heritability and generally outperform local-only modelling. Furthermore, across all simulation settings, we observed that MOSTWAS showed greater or nearly equal power to detect gene-trait associations compared to local-only models. We found that, under the setting that distal-eQTLs contribute to trait heritability, the best MOSTWAS model had greater power to detect gene-trait associations than the local-only models, with the advantage in power over local-only models increasing with increased distal expression heritability (**Fig 2A**). In comparison to BGW-TWAS, across most simulation parameters, we find similar TWAS power between BGW-TWAS and MOSTWAS. We observed slight advantages (**Figs 2A** and **S1A**) for BGW-TWAS at low trait heritability ($h_p^2 = 0.20$), higher distal expression heritability ($h_{trans}^2 = 0.25$), and higher causal eQTL proportion ($p_C = 0.20$). At the same trait and distal expression heritability settings but causal proportion $p_c = 0.01$, we find a small power advantage for MOSTWAS over BGW-TWAS (**S1A Fig**). As trait heritability increases, the positive difference in TWAS power for MOSTWAS and BGW-TWAS models over local-only models decreases slightly. Similarly, we found that as the proportion of total expression heritability attributed to distal variation increased, the positive difference in predictive performance between the best MOSTWAS model and the local-only model increased (**S2 Fig**). Under the "null" case that distal variation influences expression only in the reference panel, we observed that local-only, BGW-TWAS, and MOSTWAS models perform similarly. Only at low trait heritability ($h_p^2 = 0.20$) did local-only and BGW-TWAS models have a small advantage in TWAS power over MOSTWAS models (**Fig 2B** and https://doi.org/10.5281/zenodo.4314067). This difference was reduced at larger

## A  Distal-eQTL in both eQTL and GWAS panel

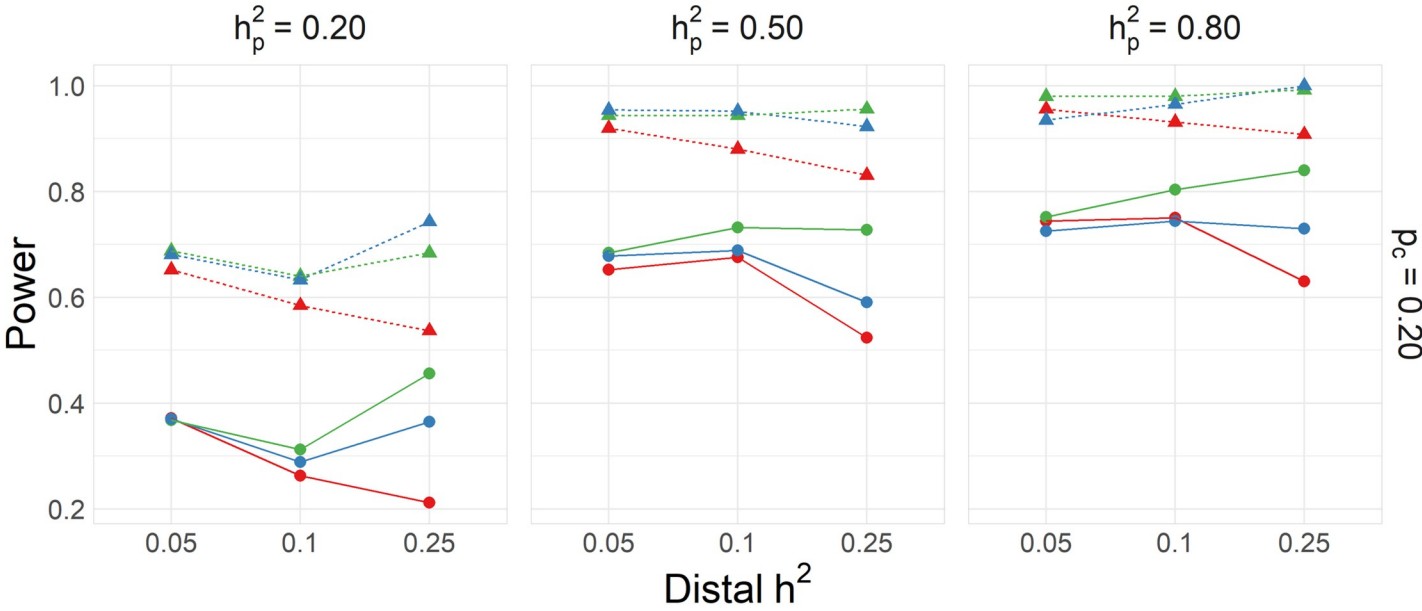

## B  Distal-eQTL only in eQTL panel

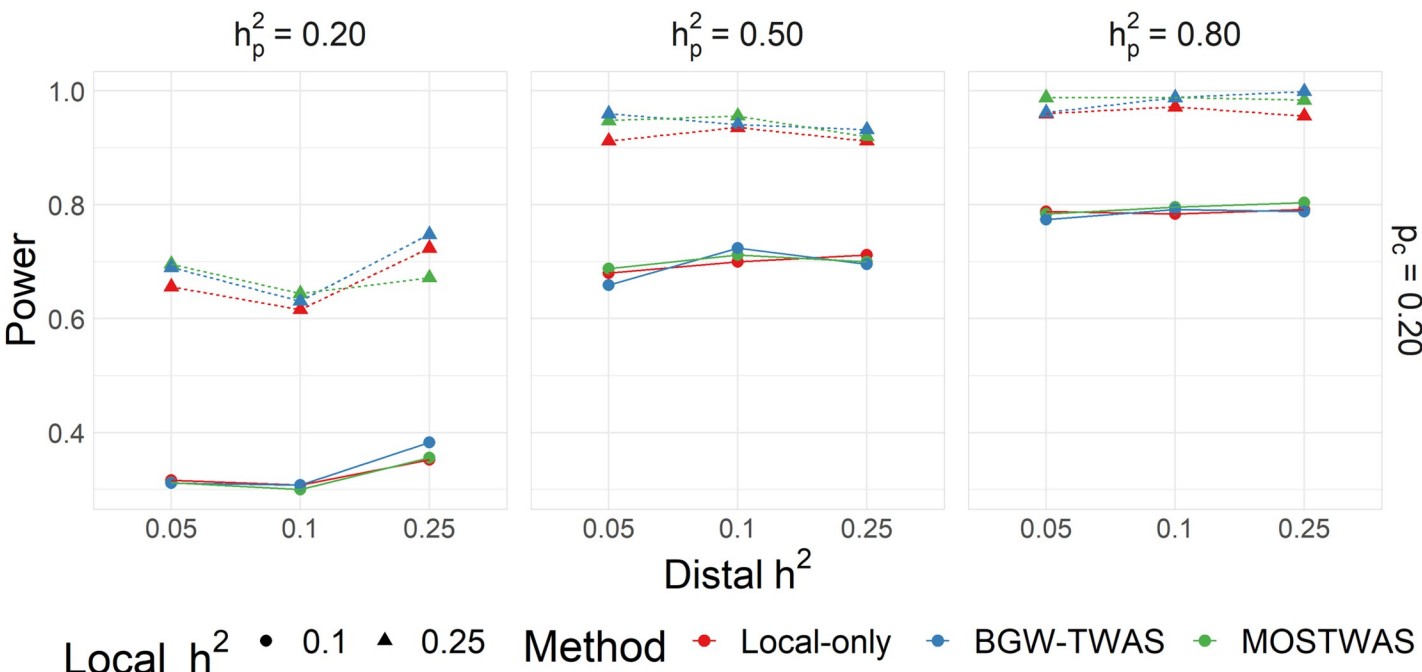

**Fig 2. Comparison of TWAS power via simulations using MOSTWAS, BGW-TWAS, and local-only models.** (A) Proportion of gene-trait associations at $P<2.5\times10^{-6}$ using local-only (red), BGW-TWAS (blue), and the most predictive MOSTWAS (green) models across various local and distal expression heritability, trait heritability, and one setting of causal eQTL proportion. (B) Proportion of significant gene-trait associations across the same simulation parameters with no distal effect on the trait in the simulated external GWAS panel.

causal proportions and trait heritability (**Fig 2B**). Using these same simulation parameters, we also simulated the false positive rate (FPR), defined as the proportion of positive associations at $P<0.05$ under the null, where the phenotype trait in the GWAS panel was permuted 1,000

times across 20 sets of simulations. We found that the FPR was generally around 0.05 for both local-only and MOSTWAS methods (**S3 Fig**).

The power of the distal-SNPs added-last test increased significantly as both the sample sizes of the eQTL reference panel and the GWAS imputation panel increased (**S4 Fig**). At a sample size of 10,000 in the GWAS panel with summary statistics (a suitably large GWAS) and a sample size greater than 200 in the eQTL panel, MOSTWAS obtained over 65% power to detect significant distal significant associations (**S4 Fig**). Overall, these results demonstrated the advantages of MOSTWAS methods for modeling the complex genetic architecture of transcriptomes, especially when distal variation has a large effect on the heritability of both the gene and trait of interest. Simulation results are provided at https://doi.org/10.5281/zenodo.4314067 [32]. The MOSTWAS package also contains functions for replicating this simulation framework.

## Real data applications in brain tissue

We applied MOSTWAS to multi-omic data derived from samples of prefrontal cortex, a tissue that has been used previously in studying neuropsychiatric traits and disorders with TWAS [33,34]. There is ample evidence from studies of brain tissue, especially the prefrontal cortex, that non-coding variants may regulate distal genes [33,35,36]; in fact, an eQTL analysis by Sng *et al* found that approximately 20–40% of detected eQTLs in the frontal cortex can be considered *trans*-acting [37]. Thus, the prefrontal cortex, in the context of neuropsychiatric disorders, provides a prime example to assess MOSTWAS.

Using ROS/MAP data on germline SNPs, mRNA expression, CpG DNA methylation, and miRNA expression ($N = 370$), we trained MeTWAS, DePMA, and local-only (FUSION without BSLMM) predictive models for the expression of all genes with significant non-zero heritability. Estimates of gene expression heritability were considerably larger when we considered distal variation with MOSTWAS (**S1 Table**). We also found that MeTWAS and DePMA performed better in cross-validation $R^2$ than local-only models (**Fig 3A, 3B and 3C**). Overall, we trained 1,385 significant local-only models, 2,287 MeTWAS models, and 4,725 DePMA models. Comparing genes with significant models with at least one of the local-only method, MeTWAS, or DePMA, we found that MeTWAS and DePMA overwhelmingly outperformed the local-only model; 86% and 79% of MeTWAS and DePMA models, respectively, had higher CV $R^2$ than corresponding local-only models. Likewise, DePMA models generally outperformed MeTWAS models, with 70% of DePMA models outperforming corresponding MeTWAS models. As shown in **Fig 4A**, median predictive $R^2$ for significant local-only models was 0.021 (25% to 75% inter-quartile interval (0.010,0.060)), for MeTWAS models was 0.030 (0.019, 0.071), and for DePMA models was 0.023 (0.017, 0.044).

We used 87 samples in ROS/MAP with genotype and mRNA expression data that were not used in model training to test portability of MOSTWAS models in independent cohorts. As shown in **Fig 4A**, DePMA and MeTWAS models obtained similar predictive adjusted $R^2$ in the external cohort: median external $R^2$ for DePMA was 0.024 (25% quantile 0.008, 75% quantile 0.035) and median $R^2$ for MeTWAS was 0.025 (0.008, 0.043) both outperforming local-only models (0.014 (0.006, 0.025)). Overall, among models with cross-validation adjusted $R^2 \geq 0.01$, 587 genes achieved external predictive $R^2 \geq 0.01$ using local-only models, 1,646 using MeTWAS, and 3,289 using DePMA.

We also compared MOSTWAS external predictive performance with BGW-TWAS, which employs Bayesian variable selection to train predictive models using both local- and distal-eQTLs [31]. Here, as BGW-TWAS models were trained on the entire ROS/MAP dataset, we extracted expression data from the PsychENCODE project for 646 samples from dorsolateral

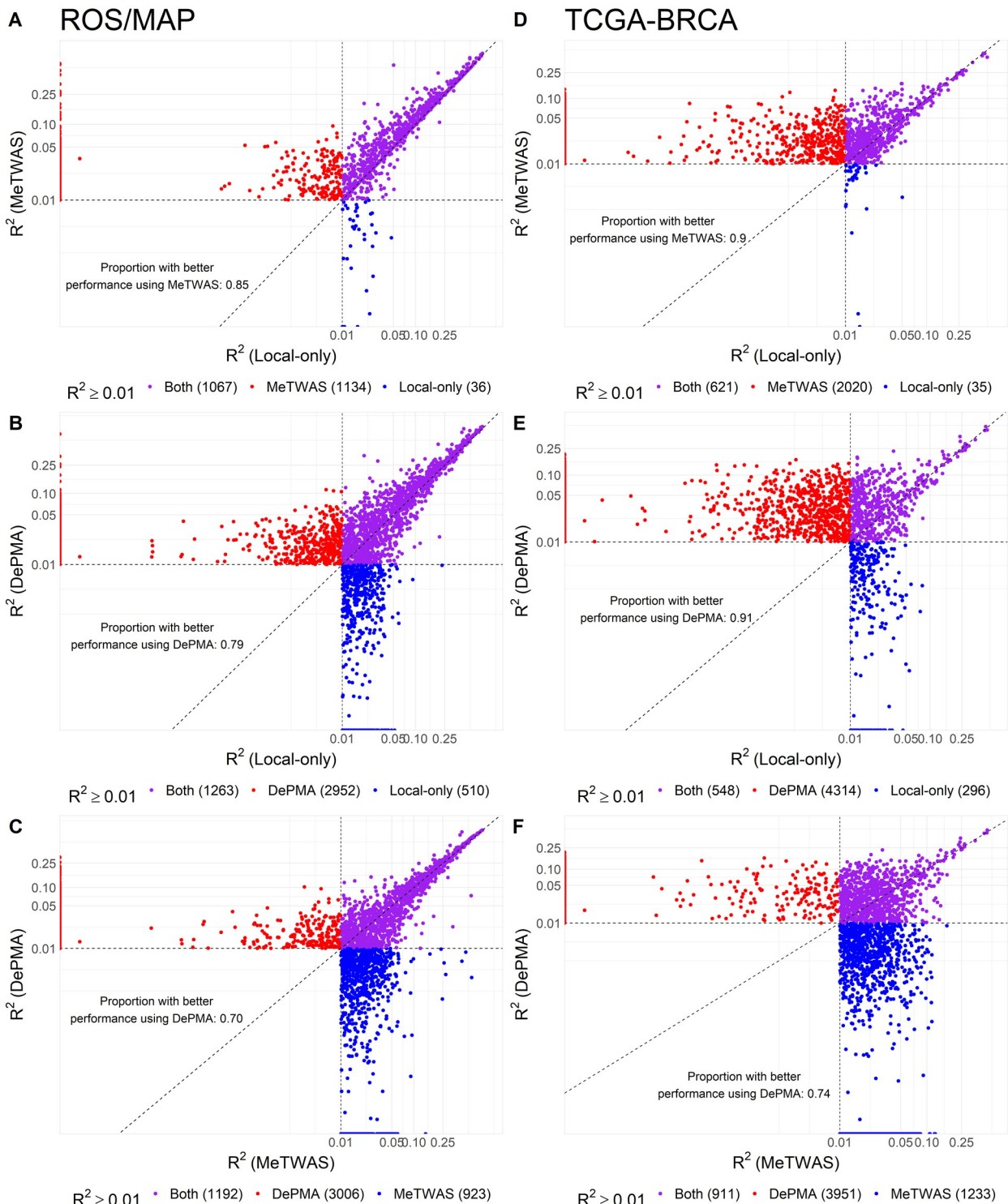

**Fig 3. Predictive adjusted $R^2$ from cross-validation across local-only, MeTWAS, and DePMA models.** If a given gene does not have $h^2 > 0$ with $P < 0.05$, we set the predictive adjusted $R^2$ to 0 here for comparison. The top row compares local-only and MeTWAS, middle row compares local-only and DePMA, and the bottom row compares MeTWAS and DePMA. The left column has performance in ROS/MAP, while the right column has performance in TCGA-BRCA. All axes indicate the CV adjusted $R^2$ for different models. We also provide the proportion of models with larger CV $R^2$ with the model reflected in the Y-axis compared to that on the X-axis.

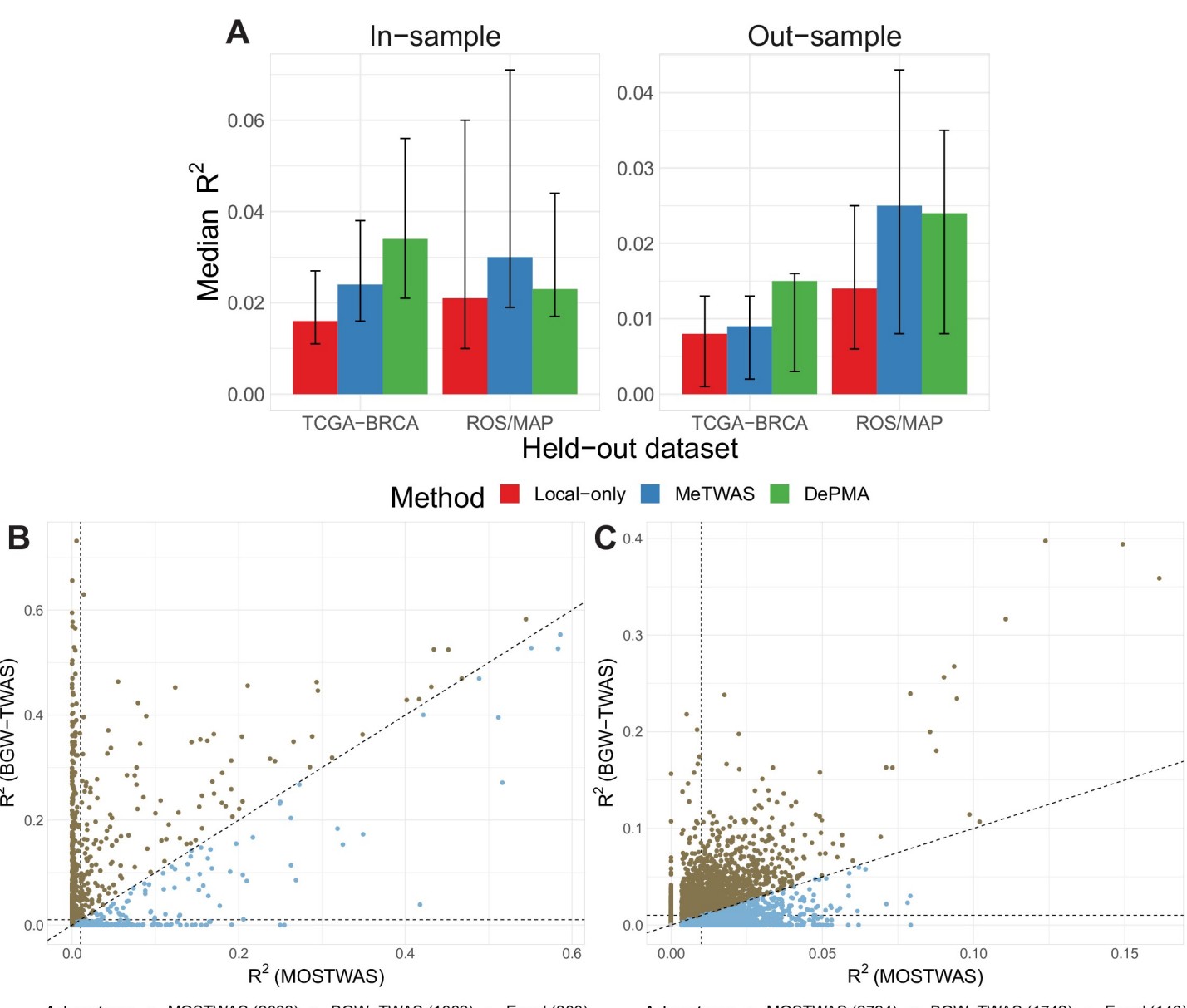

**Fig 4. Comparison of MOSTWAS prediction with other TWAS methods.** (A) Median predictive adjusted $R^2$ from in-sample and out-sample predictions using TCGA-BRCA and ROS/MAP data using local-only (red), MeTWAS (blue) and DePMA (green) models. The error bars show the 25% and 75% quantiles for predictive $R^2$ across all genes. (B) Scatterplot of predictive adjusted $R^2$ of genes using MOSTWAS (X-axis) and BGW-TWAS (Y-axis) models in a 646-sample dataset from the PsychENCODE project. The vertical and horizontal dotted lines provide reference for $R^2 = 0.01$ and the diagonal line is the 45-degree line. Each point is colored blue if the gene has a higher $R^2$ using the MOSTWAS model, gold if the gene has a higher $R^2$ with BGW-TWAS, and grey if the $R^2$ are equal. (C) Scatterplot of predictive adjusted $R^2$ of genes using MOSTWAS (X-axis) and BGW-TWAS (Y-axis) models in a 351-sample external dataset from the TCGA-BRCA. The vertical and horizontal dotted lines provide reference for $R^2 = 0.01$ and the diagonal line is the 45-degree line. Each point is colored blue if the gene has a higher $R^2$ using the MOSTWAS model, gold if the gene has a higher $R^2$ with BGW-TWAS, and grey if the $R^2$ are equal.

prefrontal cortex tissue and corresponding individual-level genotype data; these 646 samples are derived from patients in the control group, with no symptoms of schizophrenia or bipolar disorder [38,39]. We then imputed expression using both MOSTWAS and BGW-TWAS models trained using ROS/MAP and assessed predictive performance via McNemar's adjusted $R^2$. Here, we note that BGW-TWAS uses a different inclusion criteria than MOSTWAS, where a

BGW-TWAS model is trained if at least one eQTL with non-zero effect and sufficiently large posterior causal probability is detected without heritability estimation or cross-validation performance [31]. Possibly due to MOSTWAS's more conservative inclusion criteria, BGW-TWAS imputes a larger number of genes than MOSTWAS (11,990 versus 4,931, respectively). However, of the 3,385 genes that both MOSTWAS and BGW-TWAS imputed, MOSTWAS models generally have larger adjusted $R^2$ in the PsychENCODE data, where 59% of genes have higher $R^2$ with MOSTWAS compared to BGW-TWAS (**Fig 4B**). Across all genes imputable by each method, MOSTWAS and BGW-TWAS have roughly the same proportion of predictions at external validation $R^2 \geq 0.01$ (MOSTWAS 13.6%, BGW-TWAS 14.3%).

We next conducted association tests for known Alzheimer's disease risk loci using two local-only methods (PrediXcan [3] and TIGAR [40]) and the best MOSTWAS model (selected by comparing MeTWAS and DePMA cross-validation $R^2$) trained in ROS/MAP and summary-level GWAS data from the International Genomics of Alzheimer's Project (IGAP) [41]. This comparison is similar to an analysis of recapitulation of GWAS signals in Alzheimer's disease from Nagpal *et al* [40]. From literature, we identified 14 known common and rare loci of late-onset Alzheimer's disease that have been mapped to genes [41–44], all of which had MOSTWAS models with cross-validation $R^2 \geq 0.01$. (**S5 Fig**) Eight of these loci (*AKAP*, *APOE*, *CLU*, *FERMT2*, *MEF2C*, *PLCG2*, *SORL1*, *ZCWPW1*) showed significant association at nominal $P < 0.05$ (**S2 Table**), compared to those identified by PrediXcan [3] and TIGAR [40] in **Fig 5A**. MOSTWAS showed stronger associations at 8 of these loci than both local-only and DPR models. We followed up on the 8 significantly associated loci using the permutation and added-last tests (**Methods** and **S1 Text**). Four of these loci (*AKAP9*, *APOE*, *SORL1*, *ZCWPW1*) showed significant associations, conditional on variants with large GWAS effect sizes (permutation test significant at FDR-adjusted $P < 0.05$). Three of the 4 loci also showed significant associations with distal variants, above and beyond the association with local variants, at FDR-adjusted $P < 0.05$ (**S2 Table**). We also assessed TWAS associations for Alzheimer's disease risk using MOSTWAS for the 11 autosomal risk genes identified by Luningham *et al* using BGW-TWAS [31]. Due to more conservative inclusion criteria in MOSTWAS, we only trained significant models for 5 of these genes (*AEP1*, *BTN3A2*, *GPX1*, *APOC1*, and *HLA-DRB1*). At $P < 2.5 \times 10^{-6}$, we found significant associations with risk of Alzheimer's disease for *APOC1* ($Z = 4.84$, $P = 1.29 \times 10^{-6}$) and *HLA-DRB1* ($Z = 6.80$, $P = 1.05 \times 10^{-11}$) that also passed permutation testing. Overall, across all 5,407 genes with significant MOSTWAS models, 18 genes showed TWAS associations at $P < 2.5 \times 10^{-6}$ and passed permutation testing at nominal $P < 0.05$ (**Fig 5B**, **S3 Table**, **S4 Fig**).

Using the final suite of 5,407 MOSTWAS models, we also conducted a transcriptome-wide association study for risk of major depressive disorder (MDD) using summary statistics from the Psychiatric Genomics Consortium (PGC) genome-wide meta-analysis that excluded data from the UK Biobank and 23andMe [45]. Of the 618 genes that had cross-validation $R^2 \geq 0.01$ in ROS/MAP using both local-only and MOSTWAS models and non-zero SNP intersection with PGC summary statistics, we found 4 genes with significant MDD associations using both MOSTWAS and local-only models, 1 significant gene using only the MOSTWAS model, and 4 significant genes with the local-only model. Furthermore, 552 of these genes showed larger or equal strengths of association with survival using the MOSTWAS model than the local-only model (**S6 Fig**). QQ-plots for TWAS $Z$-statistics and $P$-values are provided in **S7 Fig** and **S8 Fig** for both local-only and MOSTWAS models, showing earlier departure from the null using local-only models compared to MOSTWAS. Overall, we identified 88 MDD risk-associated loci at $P < 2.5 \times 10^{-6}$, a Bonferroni correction over 20,000 tests generally used in TWAS to signify transcriptome-wide statistical significance. Of these 88 genes, 43 persisted when subjected to permutation testing at nominal $P < 0.05$ (colored red in **Fig 5C**).

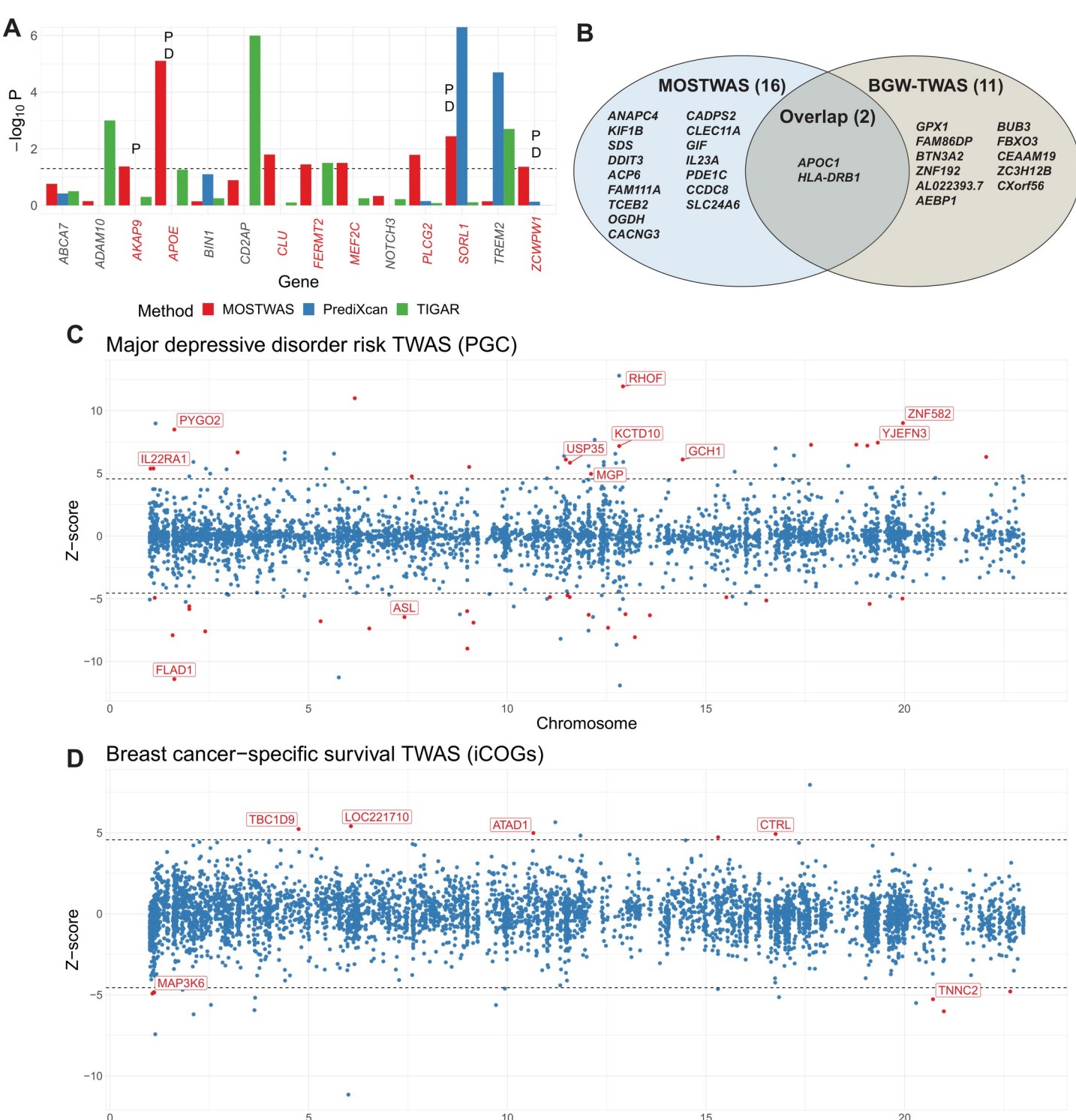

**Fig 5. TWAS results using MOSTWAS models. (A)** TWAS associations (nominal −log₁₀ P on Y-axis) with 14 known Alzheimer's risk loci (X-axis), as identified in literature, using MOSTWAS (red), PrediXcan (blue), and TIGAR Dirichlet process regression (green). Loci are labeled with P if the permutation test achieves $P<0.05$ and D if the added-last test achieves Bonferroni-adjusted $P<0.05$. **(B)** Putative TWAS genes identified by MOSTWAS (right) and BGW-TWAS (right). Genes specific to each method and in common are labelled. **(C)** TWAS associations for major depressive disorder risk using GWAS summary statistics from PGC. Loci are colored red if the overall association achieves FDR-adjusted $P<0.05$ and the permutation test also achieves FDR-adjusted $P<0.05$. We label the 7 loci that were independently validated with UK Biobank GWAX summary statistics at FDR-adjusted $P<0.05$ for both the overall association test and permutation test. **(D)** TWAS associations for breast cancer-specific survival using GWAS summary statistics from iCOGs. Loci are colored red and labeled if the overall association achieves FDR-adjusted $P<0.05$.

Lastly, we downloaded genome-wide association study by proxy (GWAX) summary statistics from the UK Biobank [46] for replication analysis of loci identified using PGC summary statistics. We found 11 of these 43 loci (labeled in **Fig 4C** and listed in **S4 Table**) also showed an association in UK Biobank GWAX that was in the same direction as in PGC. In comparison, using local-only models, we identified 5 genes associated with MDD risk at $P<2.5\times10^{-6}$ that persisted after permutation testing; one of these loci showed transcriptome-wide significant associations in the UK Biobank GWAX in the same direction as in PGC (*ZKSCAN3*, $Z = 10.67$ in PGC; $Z = 5.44$ in UK Biobank GWAX). These replication rates between MOSTWAS and local-only models were similar (accounting for the total number of associations), highlighting that the inclusion of distal variation does not hinder the replicability of MOSTWAS associations in comparison to local-only models [46,47]. TWAS results are provided at https://doi.org/10.5281/zenodo.4314067. It is important to note here that the UK Biobank dataset is not a GWAS dataset as it defined a case of MDD as any subject who has the disorder or a first-degree relative with MDD. Hence, the study forfeits study power to detect gene-trait associations for MDD [46,47]. Nonetheless, we believe that strong prediction in independent cohorts and TWAS results across two independent cohorts provided an example of the robustness of MOSTWAS models.

In summary, we observed that MOSTWAS models generally had higher predictive $R^2$ than local-only models both in training and independent cohorts. In addition, we observed that, for genes that could be imputed by both MOSTWAS and BGW-TWAS, MOSTWAS models outperformed BGW-TWAS models in external datasets, though BGW-TWAS trained more significant models, possibly due to differences in inclusion criteria between the two methods. We also found that MOSTWAS recapitulated 8 known Alzheimer's risk loci that were not detected by local-only modeling (both PrediXcan [3] and TIGAR [40]), 3 of which had significant distal associations above and beyond the information in local variants using our added-last test. We also illustrated that some MDD-risk-associated loci detected by MOSTWAS in a GWAS cohort were replicable in an independent GWAX cohort [45,46].

## Real data applications in breast cancer tumors

We applied MOSTWAS using breast tumor multi-omics and disease outcomes, motivated by recent GWAS and TWAS for breast cancer-specific survival [48–52]. Previous breast tumor eQTL studies have revealed several significant distal-eQTLs in trait-associated loci, many of which are in regulatory or epigenetic hotspots [52,53], motivating our application of MOSTWAS in breast tumor expression modeling.

Using TCGA-BRCA [24] datasets for germline SNPs, tumor mRNA expression, CpG DNA methylation, and miRNA expression ($N = 563$), we trained MeTWAS, DePMA, and local-only (FUSION without BSLMM) predictive models for the mRNA expression of all genes with significant non-zero germline heritability at $P<0.05$. Estimates of heritability for genes were 2–5 times larger when we considered distal variation using MOSTWAS methods (**S1 Table**). We also found that MeTWAS and DePMA performed better in cross-validation $R^2$, with larger numbers of models at $R^2\geq0.01$ and significant germline heritability using MOSTWAS models than local-only models (**Fig 3D, 3E and 3F**). In total, we trained 849 significant local-only models, 2,641 significant MeTWAS models, and 4,862 significant DePMA models. Of these significant models, median predictive $R^2$ for local-only models was 0.016 (25% to 75% interquartile interval (0.011,0.027)), for MeTWAS models was 0.024 (0.0161, 0.038), and for DePMA models was 0.034 (0.021, 0.056). In addition to cross-validation, we used 351 samples in TCGA-BRCA with only genotype and mRNA expression data, which were not used in model training, to test the portability of MOSTWAS models in independent external cohorts.

As shown in **Fig 4A**, DePMA models obtained the highest predictive adjusted $R^2$ in the external cohort (median 0.014, 25% to 75% inter-quartile interval (0.003,0.016)), with MeTWAS models (0.009, (0.002,0.013)) performing on par with local-only models (0.008, (0.001, 0.013)), considering only genes that showed significant heritability and cross-validation adjusted $R^2{\geq}0.01$ using a given method. Overall, among genes with cross-validation adjusted $R^2{\geq}0.01$, 153 achieved external predictive $R^2{\geq}0.01$ using local-only models, 449 using MeTWAS, and 2,527 using DePMA.

We also trained expression models with BGW-TWAS using TCGA-BRCA data, computed cross-validation adjusted $R^2$ across three folds, and imputed expression into this external dataset for comparison. We trained 14,029 models using BGW-TWAS at default input parameters with at least one SNP in the model, compared to 5,897 models using MOSTWAS. Over 3-fold cross-validation, we found that 6.2% of these models met $R^2{>}0.01$. Comparing the 10,202 genes with BGW-TWAS models and $h^2{>}0$ with $P{<}0.05$, as calculated in the MOSTWAS pipeline, we found that 77% of MOSTWAS models outperform BGW-TWAS models in cross-validation (**S9 Fig**). Of the 4,682 genes that passed both MOSTWAS and BGW-TWAS inclusion criteria, MOSTWAS models generally had larger adjusted $R^2$ in the external dataset, whereas nearly 60% of genes had higher $R^2$ with MOSTWAS compared to BGW-TWAS (**Fig 4C**). Across all genes with models of each method, MOSTWAS (5,897 significant models) trained more models with external validation $R^2{\geq}0.01$ (MOSTWAS 46.9%, BGW-TWAS 38.5%).

Lastly, we conducted association studies for breast cancer-specific survival using local-only and the MOSTWAS model with largest $R^2$ trained in TCGA-BRCA and summary-level GWAS data from iCOGs [51]. Here, we constructed the weighted burden test, as described above and in Pasaniuc *et al* and Gusev *et al* [4,29]. We prioritized genes with $P{<}2.5{\times}10^{-6}$ (Bonferroni correction across 20,000 genes) for permutation testing. Of the 377 genes that had cross-validation $R^2{\geq}0.01$ in TCGA-BRCA using both local-only and MOSTWAS models and non-zero SNP intersection with iCOGs summary statistics, we found no transcriptome-wide significant survival associations with the same loci. Furthermore, 370 of these loci showed larger or equal strengths of association with survival using the MOSTWAS model than the local-only model (**S6 Fig**). QQ-plots for TWAS $Z$-statistics (**S7 Fig**) and $P$-values (**S8 Fig**) showed earlier departure from the null using local-only models. These results in TCGA-BRCA demonstrated the improved transcriptomic prediction and power to detect gene-trait associations using MOSTWAS over local-only modeling.

**Functional hypothesis generation with MOSTWAS.** We next conducted TWAS for breast cancer-specific survival using all genes with significant germline heritability at $P{<}0.05$ with the most predictive MOSTWAS model (i.e. MeTWAS or DePMA model with the larger cross-validation $R^2$ greater than 0.01); this final set of models included 5,897 genes. We identified 23 survival-associated loci at $P{<}2.5{\times}10^{-6}$. Of these 23 loci, 11 persisted when subjected to permutation testing at nominal $P{<}0.05$ (colored red in **Fig 5D** and **S5 Table**).

An advantage of MOSTWAS is its ability to aid in functional hypothesis generation for mechanistic follow-up studies. The distal-SNP added-last test allows for identification of genes where trait association from distal variation is significant, above and beyond the contribution of the local component. For 6 of the TWAS-associated 11 loci, at FDR-adjusted $P{<}0.05$, we found significant distal variation added-last associations (see **S1 Text** and **S5 Table**), suggesting that distal variation may contribute to the gene-trait associations. These loci showed distal association with the gene of interest mediated through transcription factors and methylation sites, many of which have critical roles in breast cancer proliferation. For example, we found that *ATAD1*, *TBC1D9*, and *TNNC2* expression are all mediated by transcription factors *ACTRT2*, *MMP23B*, and *TP73* and methylation sites around *MMP23B* and *TP73*; these transcription factors have known functions in tumor suppression or proliferation [54–57]. In fact,

it has been shown that both *MMP23B* and *TP73* are affected by hypermethylation of CpG sites in their promoter region, a mechanism that may activate breast cancer tumorigenesis [58,59]. We also found two transcription factors (*ROCK2*, *USF3*) in the distal components of the *CTRL-* and *MAP3K6*-associations with survival; both transcription factors are interconnected within the MAPK pathway, known to be involved in breast cancer proliferation [60–63]. These regulatory sites serve as an example of how distal genomic regions can be prioritized for functional follow-up studies to elucidate the mechanisms underlying the SNP-gene-trait associations. These results showed the strength of MOSTWAS to detect and prioritize gene-trait associations that are influenced by distal variation and to aid in generating functional hypotheses for these distal relationships.

**Differences in eQTL architecture leveraged by MeTWAS and DePMA.**    To study how eQTL architecture differs across genes well-predicted by MeTWAS and/or DePMA, we looked at genome-wide eQTLs of three genes: *LQK1* (high CV $R^2$ with both MeTWAS and DePMA), *C6orf94* (high $R^2$ only with MeTWAS), and *TSC22D4* (high $R^2$ only with DePMA), with Manhattan plots of eQTL associations in **S10 Fig**. Here, we saw that *LQK1* bears a strong local-eQTL signal that is leveraged by both models. In *C6orf94*, we find that eQTL signal is distributed evenly across the genome and the superior MeTWAS model leverages clusters of distal-eQTLs that are local to gene-associated mediators. In contrast, the distal-eQTL signal for *TSC22D4* is concentrated in a single peak with linkage disequilibrium (LD) support. This eQTL analysis provides some insight into genetic architectures that favor a MOSTWAS method over the other.

## Comparison of computation time

To assess the difference in computational burden between local-only, MeTWAS, and DePMA modeling, we randomly selected a set of 50 genes that are heritable across all three models from TCGA-BRCA and computed per-gene time for fitting models serially using a high-performance cluster (RedHat Debian operating system) with 3.0 GHz processor and 8 gigabytes of RAM. As shown in **S11 Fig**, we found that MeTWAS (average of 40 seconds per gene) and DePMA (average 193 seconds per gene) took approximately 1–6 times longer to fit than a traditional FUSION-like local-only model (average 36 seconds) [4]. These computation times do not include QTL detection steps (detecting distal mediators in MeTWAS and distal-eQTLs in DePMA) prior to fitting the predictive models. The computational bottlenecks for MeTWAS and DePMA are fitting mediator prediction models and testing the mediation effect through permutation, respectively. Model-fitting here includes heritability estimation, estimating the SNP-expression weights, and cross-validation. MeTWAS and DePMA are aided considerably by the efficient memory mapping of the bigsnpr package and the in-built LASSO implementation from the biglasso package [64,65]; presumably, local-only modeling using the trainLocalModel() function in MOSTWAS will also speed up training FUSION-like models for cis-only TWAS (without Bayesian sparse linear mixed modeling). We have also implemented parallelized methods to train an expression model for a single gene in MOSTWAS. We recommend fitting an entire set of genes from an RNA-seq panel via a batch computing approach [66–68]. Using a parallel implementation with 5 cores and batch computing, we trained MOSTWAS expression models for 15,568 genes from TCGA-BRCA in approximately 21 hours.

## Discussion

Through simulation analysis and real applications using two datasets [23,24], we demonstrated that multi-omic methods that prioritize distal variation in TWAS have higher predictive performance and power to detect tissue-specific gene-trait associations [9,13,69], especially when

distal variation contributes substantially to trait heritability. We proposed two methods (MeT-WAS and DePMA) for identifying and including distal genetic variants in gene expression prediction models. Specifically, our methods recover traditional local-only TWAS models and associations when distal genetic variants have little influence on the heritability of gene expression [4]; however, when heritable mediators or distal-eQTLs explain variance in gene expression, local-only models ignore this important information that MOSTWAS has the ability to leverage. MeTWAS, a two-step regression method that leverages SNPs local to gene-associated mediators that are heritable, is preferred for genes that are influenced by a complex network of molecular features across the genome. A tissue-wide approach that searches for distal eQTL signals conserved across different tissues could be an extension for MeTWAS [70]. In contrast, DePMA, which prioritizes significant distal-eQTLs with large mediation effects on the gene, may perform better when distal eQTLs that have strong LD support and are local to regulatory machinery that have indirect effects on transcription of a gene. These distal eQTLs with large mediated effects may be tissue-specific, as they generally cluster around epigenomic or regulatory markers [71–73]. We have provided implementations of these methods in MOSTWAS (Multi-Omic Strategies for Transcriptome-Wide Association Studies), an R package available freely on GitHub.

Not only does MOSTWAS improve transcriptomic imputation both in- and out-of-sample, but it also provides a test for the identification of heritable mediators that affect eventual transcription of the gene of interest. These identified mediators can provide insight into the underlying mechanisms for SNP-gene-trait associations to improve detection of gene-trait associations and to prioritize biological units for functional follow-up studies. TWAS using MOSTWAS models was able to recapitulate 8 out of 14 known Alzheimer's disease risk loci in IGAP GWAS summary statistics [41], many of which were not recoverable with local-only models. We showed the utility of the distal-SNPs added-last test to prioritize significant distal SNP-gene-trait associations for follow-up mechanistic studies, which could not be identified using traditional local-only TWAS. In PGC GWAS summary-level data for major depressive disorder [45], we found 43 risk loci, 11 of which were replicated in independent GWAX summary statistics from the UK Biobank [46]. A few of these genes have been implicated in neuro-psychiatric traits or disorders. For example, genetic and methylomic variation in *YJEFN3* has shown associations with schizophrenia [74,75] and common and rare genetic variants in *GCH1* are associated with Parkinson's disease [76]. Furthermore, *KCTD10* encodes a protein that belongs to a family implicated in fetal development of many psychiatric traits [77,78]. Using MOSTWAS and iCOGs summary-level GWAS statistics for breast cancer-specific survival [51], we identified 6 survival-associated loci involved with p53 binding and oxidoreductase activity pathways [79,80]. These loci include *MAP3K6*, which encodes a mitogen-activated protein kinase, a signaling transduction molecule involved in the progression of aggressive breast cancer subtypes [81]. The utility of the distal-SNPs added-last test was highlighted in the breast-cancer survival TWAS, where we uncovered a set of transcription factors and methylation sites predicted to mediate expression of these survival-associated genes. Many of these mediators (e.g. *MMP23D*, *TP73*, *ROCK2*, *USF3*) have been implicated in tumorigenesis and tumor suppression or progression pathways [54–63]. None of the risk- or survival-associated loci identified by MOSTWAS were detected using local-only models.

We compared MOSTWAS to a contemporary TWAS method that incorporates distal-eQTLs, BGW-TWAS [31]. BGW-TWAS can impute a larger number of gene models than MOSTWAS, but among genes both BGW-TWAS and MOSTWAS were able to impute in external PsychENCODE and TCGA-BRCA data [38,39], MOSTWAS generally showed larger predictive adjusted $R^2$. MOSTWAS's stricter inclusion criterion (thresholds for both expression heritability and cross-validation performance) may lead to fewer genes trained and

ultimately tested; for example, when we subjected BGW-TWAS models trained with TCGA-BRCA data to cross-validation, few met MOSTWAS's inclusion criterion of cross-validation adjusted $R^2 > 0.01$. We believe that assessing expression heritability signal and cross-validating models prior to TWAS is necessary: as Ding *et al* points out in their systematic power analysis, expression heritability is a major factor in determining TWAS power [82]. Alternative methods of fitting the predictive model, like TIGAR's Dirichlet Prior Regression, could be built into the MOSTWAS framework to boost predictive power [40]. In a TWAS for Alzheimer's disease, results from MOSTWAS and BGW-TWAS showed two overlap genes, out of a total of 18 and 13 putative TWAS genes in each respective method. BGW-TWAS employs Bayesian methodology to scan the entire genome for potentially causal distal-eQTLs and can build models for more genes at a large computational cost [31]. For example, BGW-TWAS requires approximately 30 minutes per gene with 3 GB of memory with parallel computation over 4 cores, whereas DePMA (the slower of the two MOSTWAS methods) takes approximately 6 minutes per gene with 8 GB of memory with serial implementation. Note that BGW-TWAS computation time includes genome-wide single-variant eQTL tests. BGW-TWAS only requires genetic and transcriptomic data for its prediction, an advantage over MOSTWAS. On the other hand, MOSTWAS has an interpretational advantage over BGW-TWAS by pinning distal variants to mediating biomarkers and testing trait association at these distal loci with the added-last test.

A limitation of MOSTWAS is the increased computational burden over local-only modeling, especially in DePMA's permutation-based mediation analysis for multiple genome-wide mediators. By making some standard distributional assumptions on the SNP-mediator effect size and mediator-gene effect size vectors (e.g. effect sizes following a multivariate Normal distribution with non-zero off-diagonal covariance), we believe a Monte-Carlo resampling method to estimate the null distribution of the product of these two effect size vectors may decrease computational time without significant loss in statistical power [83]. Nevertheless, we believe that MOSTWAS's gain in predictive performance and power to detect gene-trait associations outweighs the added computational cost, especially with our implementation leveraging the bigsnpr and bigstatsr packages [64]. Furthermore, compared to BGW-TWAS [31], model training in MOSTWAS is considerably faster. Another concern with the inclusion of distal variants is that RNA-sequencing alignment errors can lead to false positives in distal-eQTL detection [84], and in turn, bias the mediation modeling. Cross-mapping estimation, as described by Saha *et al*, can be used to flag potential false positive distal-QTLs that are detected in the first step of MeTWAS and DePMA. Another limitation of MOSTWAS is the general lack of rich multi-omic panels, like ROS/MAP and TCGA-BRCA, that provide a large set of mediating biomarkers that may be mechanistically involved in gene regulation. However, the two-step regression framework outlined in MeTWAS allows for importing mediator intensity models trained in other cohorts to estimate the germline portion of total gene expression from distal variants. Importing mediator models from an external cohort can also reduce the testing burden in the preliminary QTL analysis in MeTWAS and DePMA.

MOSTWAS provides a user-friendly and intuitive tool that extends transcriptomic imputation and association studies to include distal regulatory genetic variants. We demonstrate that the methods in MOSTWAS based on two-step regression and mediation analysis generally out-perform local-only models in both transcriptomic prediction and TWAS power without signs of inflated false positive rates, though at the cost of longer computation time. MOSTWAS enables users to utilize rich reference multi-omic datasets for enhanced gene mapping to better understand the genetic etiology of polygenic traits and diseases with more direct insight into functional follow-up studies.

## Methods

We first outline the two methods proposed in this work: (1) mediator-enriched TWAS (MeT-WAS) and (2) distal-eQTL prioritization via mediation analysis (DePMA). MeTWAS and DePMA are combined in the MOSTWAS R package, available at www.github.com/bhattacharya-a-bt/MOSTWAS. MOSTWAS employs the bigstatsr and bigsnpr packages for efficient memory mapping and faster computation [64]. Full mathematical details are provided in **S1 Text**.

### Transcriptomic prediction using MeTWAS

Across all samples in the training dataset and for a single gene of interest, MeTWAS, an adaptation of two-step regression, takes in a vector of gene expression, the matrix of genotype dosages local to the gene of interest (default of 1 Megabase around the gene), and a set of mediating biomarkers that are estimated to be significantly associated with the expression of the gene of interest through a QTL analysis. In accordance with previous studies that use penalized regression methods [52,85,86], we only select the most significant gene-associated mediators as adding too many potentially redundant features often leads to poorer predictive performance. This feature selections also limits computational time. Through simulations, we observed that including all SNPs local to the mediators results in lower predictive $R^2$ compared to the two-step regression method in MeTWAS (**S12 Fig**). These mediating biomarkers can be DNA methylation sites, microRNAs, transcription factors, or any molecular feature that may be genetically heritable and affect transcription.

Transcriptome prediction in MeTWAS draws from two-step regression, as summarized in **Fig 1A**. Using the genotype local to these mediators, MeTWAS first trains a predictive model for their intensities (i.e. expression, methylation, etc.) using either elastic net [26] or linear mixed modeling [27]. In practice, we found that a simpler, one-step procedure of including all variants local to both the gene and to potential mediators led to the distal SNP effects being estimated as zero during the regularization process, even in simulations when the true distal SNP effects were nonzero. We then use these predictive models to estimate the genetically regulated intensity (GRIn) of each mediator in the training set, through cross-validation. The GRIns for each mediator are then included in a matrix of fixed effects. The effect sizes of the GRIns on the expression of the gene of interest are estimated using ordinary least squares regression, and then the expression vector is residualized for these effect sizes. Effect sizes of variants local to the gene of interest are then estimated using elastic net or linear mixed modeling [26,27] on the residualized gene expression quantity. Details are provided in **S1 Text**.

### Transcriptomic prediction using DePMA

Expression prediction in DePMA hinges on prioritizing distal-eSNPs via mediation analysis for inclusion in the final DePMA predictive model, adopting methods from previous studies [11,12,14]. A multi-omic dataset with gene expression, SNP dosages, and potential mediators is first split into training-testing subsets. Based on the minor allele frequencies of SNPs and total sample size, we recommend a low number of splits (less than 5).

In the training set, we identify mediation test triplets that consist of (1) a gene of interest, (2) a distal-eSNP associated with the expression of the gene (default of $P < 10^{-6}$), and (3) a set of mediating biomarkers local to and associated with the distal-eSNP (default of FDR-adjusted $P < 0.05$). We estimate the total indirect mediation effect (TME) of the distal-eSNP on the gene of interest mediated through the set of these mediators, as defined by Sobel [87]. We assess the magnitude of this indirect effect using a two-sided permutation test to obtain a permutation $P$-value, as more direct methods of computing standard errors for the estimated TME are often

biased [14,88]. We also provide an option to estimate an asymptotic approximation to the standard error of the TME and conduct a Wald-type test. This asymptotic option is significantly faster at the cost of inflated false positives (see **S1 Text** and **S13 Fig**). Distal-eSNPs with significantly large absolute TMEs are included with the local SNPs for the gene of interest in a predictive model, fit using elastic net or linear mixed modeling [26,27]. These SNP effect sizes can then be exported for imputation in external GWAS cohorts. Details are provided in **S1 Text**.

### Transcriptomic imputation with MOSTWAS

In an external GWAS panel, if individual level genotypes are available, we construct the mediator-enriched genetically regulated expression (GReX) of the gene of interest by multiplying the genotypes in the GWAS panel by the effect sizes estimated in a MOSTWAS model. This GReX quantity represents the component of total expression that is attributed to germline genetics and can be used in downstream TWAS to detect gene-trait associations.

### Tests of association

If individual level genotypes are not available, then the weighted burden $Z$-test, proposed by Pasaniuc *et al* and employed in FUSION [4,29], can be employed and applied to summary statistics. Briefly, the test statistic is a linear combination of the $Z$-scores corresponding to the SNPs included in the MOSTWAS model for a gene of interest, where each individual GWAS $Z$-score is weighted by the corresponding MOSTWAS effect size. The covariance matrix for this weighted burden test statistic is estimated from the linkage disequilibrium between SNPs in the eQTL panel or some publicly available ancestry-matched reference panels. This weighted burden test statistic is compared to the standard Normal distribution for inference.

We implement a permutation test, conditioning on the GWAS effect sizes to assess whether the same distribution SNP effect sizes could yield a significant association by chance [4]. We permute the effect sizes 1,000 times without replacement and recompute the weighted burden test statistic to generate permutation null distribution. This permutation test is only conducted for overall associations at a user-defined significance threshold (default to FDR-adjusted $P<0.05$).

Lastly, we also implement a test to assess the information added from distal-eSNPs in the weighted burden test beyond what we find from local SNPs. This test is analogous to a group added-last test in regression analysis, applied here to GWAS summary statistics. Formally, we test whether the weighted burden test statistic for the distal-SNPs is significantly non-zero given the observed weighted burden test statistic for the local-SNPs. We draw conclusions from the assumption that these two weighted burden test statistics follow bivariate Normal distribution. Full details and derivations are given in **S1 Text**.

### Simulation framework

We first conducted simulations to assess the predictive capability and power to detect gene-trait associations under various settings for phenotype heritability ($h_p^2 \in \{0.2, 0.5, 0.8\}$), local ($h_l^2 \in \{0.1, 0.25\}$) and distal heritability ($h_d^2 \in \{0.1, 0.25\}$) of expression, and proportion of causal local and distal SNPs ($p_c \in \{0.01, 0.20\}$). We considered two scenarios for each set of simulation parameters: (1) an ideal case where the leveraged associated between the distal-SNP and gene of interest exists in both the reference and imputation panel, and (2) a "null" case where the leveraged association between the distal-SNP and the gene of interest exists in the reference panel but does not contribute phenotype heritability in the imputation panel.

Using genetic data from TCGA-BRCA as a reference, we used SNPs local to the gene *ESR1* (Chromosome 6) to generate local eQTLs and SNPs local to *FOXA1* (Chromosome 14) to generate distal-eQTLs for a 400-sample eQTL reference panel and 1,500-sample GWAS imputation panel, as in Mancuso *et al*'s *twas_sim* protocol [89]. We computed the adjusted predictive $R^2$ in the reference panel for the trained MeTWAS and DePMA models and tested the gene-trait association in the GWAS panel using the weighted burden test. The association study power was defined as the proportion of gene-trait associations with $P<2.5\times10^{-6}$, the Bonferroni-corrected significance threshold for testing 20,000 independent genes across 1,000 simulations under each set of simulation parameters. With these simulated datasets, we also assessed the power of the distal added-last test by computing the proportion of significant distal associations, conditional on the local association at FDR-adjusted $P<0.05$. Full details are provided in S1 Text.

## Data acquisition

**Multi-omic data from ROS/MAP.** We retrieved imputed genotype, RNA expression, miRNA expression, and DNA methylation data from The Religious Orders Study and Memory and Aging Project (ROS/MAP) Study for samples derived from human pre-frontal cortex [23,90,91]. We excluded variants (1) with a minor allele frequency of less than 1% based on genotype dosage, (2) that deviated significantly from Hardy-Weinberg equilibrium ($P<10^{-8}$) using appropriate functions in PLINK v1.90b3 [92,93], and (3) located on sex chromosomes. Final ROS/MAP genotype data was coded as dosages, with reference and alternative allele coding as in dbSNP. We intersected to the subset of samples assayed for genotype (at 4,141,537 variants), RNA-seq (15,857 genes), miRNA-seq (247 miRNAs), and DNA methylation (391,626 CpG sites), resulting in a total of 370 samples. Again, we only considered the autosome in our analyses. We adjusted gene and miRNA expression and DNA methylation by relevant covariates (10 principal components of the genotype age at death, sex, and smoking status).

**Genetic and gene expression data from PsychENCODE.** For external validation and comparison to BGW-TWAS, we obtained genetic and RNA-seq expression data from 646 control samples from the PsychENCODE project. Quality control and pre-processing of this data has been described previously [38,39]. Gene expression data for all 646 samples was derived from dorsolateral prefrontal cortex tissue and was residualized for the following covariates: post-mortem interval (PMI), RNA integrity number (RIN), age, sex, the square of PMI, the square of RIN, and 50 hidden covariates with prior, as estimated using Mostafavi *et al*'s methodology [94].

**Multi-omic data from TCGA-BRCA.** We retrieved genotype, RNA expression, miRNA expression, and DNA methylation data for breast cancer indications in The Cancer Genome Atlas (TCGA). Birdseed genotype files of 914 subjects were downloaded from the Genome Data Commons (GDC) legacy (GRCh37/hg19) archive. Genotype files were merged into a single binary PLINK file format (BED/FAM/BIM) and imputed using the October 2014 (v.3) release of the 1000 Genomes Project dataset as a reference panel in the standard two-stage imputation approach, using SHAPEIT v2.87 for phasing and IMPUTE v2.3.2 for imputation [95–97]. We excluded variants (1) with a minor allele frequency of less than 1% based on genotype dosage, (2) that deviated significantly from Hardy-Weinberg equilibrium ($P<10^{-8}$) using appropriate functions in PLINK v1.90b3 [92,93], and (3) located on sex chromosomes. Final TCGA genotype data was coded as dosages, with reference and alternative allele coding as in dbSNP.

TCGA level-3 normalized RNA-seq expression data, miRNA-seq expression data, and DNA methylation data collected on Illumina Infinium HumanMethylation450 BeadChip were

downloaded from the Broad Institute's GDAC Firehose (2016/1/28 analysis archive) via Fire-Browse [24,98]. We intersected to the subset of samples assayed for genotype (4,564,962 variants), RNA-seq (15,568 genes), miRNA-seq (1,046 miRNAs), and DNA methylation (485,578 CpG sites), resulting in a total of 563 samples. We only considered the autosome in our analyses. We adjusted gene and miRNA expression and DNA methylation by relevant covariates (10 genotype principal components, tumor stage at diagnosis, and age).

**Summary statistics for downstream association studies.** We conducted TWAS association tests using relevant GWAS summary statistics for breast cancer-specific survival, risk of late-onset Alzheimer's disease, and risk of major depressive disorder. We also downloaded GWAS and genome-wide association by proxy (GWAX) summary statistics for risk of major depressive disorder (MDD) from the Psychiatric Genomics Consortium [45] and the UK Biobank [46], respectively. IGAP is a large two-stage study based on GWAS on individuals of European ancestry. In stage 1, IGAP used genotyped and imputed data on 7,055,881 single nucleotide polymorphisms (SNPs) to meta-analyze four previously-published GWAS datasets consisting of 17,008 Alzheimer's disease cases and 37,154 controls (The European Alzheimer's disease Initiative–EADI, the Alzheimer Disease Genetics Consortium–ADGC, The Cohorts for Heart and Aging Research in Genomic Epidemiology consortium–CHARGE, The Genetic and Environmental Risk in AD consortium—GERAD). In stage 2, 11,632 SNPs were genotyped and tested for association in an independent set of 8,572 Alzheimer's disease cases and 11,312 controls. Finally, a meta-analysis was performed combining results from stages 1 and 2. We downloaded iCOGs GWAS summary statistics for breast cancer-specific survival for women of European ancestry [51]. All studies and funders as listed in Michailidou *et al* [49,50] and in Guo *et al* [51] are acknowledged for their contributions. Furthermore, we downloaded GWAS summary statistics for risk of late-onset Alzheimer's disease from the International Genomics of Alzheimer's Project (IGAP) [41].

## Model training and association testing in ROS/MAP and TCGA-BRCA

Using both ROS/MAP and TCGA-BRCA multi-omic data, we first identified associations between SNPs and mediators (transcription factor genes, miRNAs, and CpG methylation sites), mediators and gene expression, and SNPs and gene expression using MatrixEQTL [99]. These QTL analyses were adjusted for 10 genotype principal components to account for population stratification, along with other relevant covariates (age, sex, and smoking status for ROS/MAP; tumor stage and age for TCGA-BRCA). For MeTWAS modeling, we considered the top 5 mediators associated with the gene of interest, assessed by the smallest FDR-adjusted $P<0.05$. For DePMA models, we considered all distal-SNPs associated with gene expression at raw $P<10^{-6}$ and any local mediators at FDR-adjusted $P<0.05$. Local windows for all models were set to 0.5 Mb. For association testing, we considered only genes with significant non-zero estimated total heritability by GCTA-LDMS [28] and cross-validation adjusted $R^2 \geq 0$ across 5 folds. The MeTWAS or DePMA model with larger cross-validation $R^2$ was considered as the final MOSTWAS model for each gene. All other modeling options in MeTWAS and DePMA were set to the defaults provided by the MOSTWAS package. Local-only modeling, unless otherwise noted, is a default implementation of FUSION without considering Bayesian sparse linear mixed modelling (BSLMM), as BSLMM is computational expensive [4]. We consider default PrediXcan and DPR modeling only in the comparison of recapitulated GWAS signals in Alzheimer's disease risk [3,40]. We downloaded BGW-TWAS weights from Luningham *et al* [31] and imputed expression in the sample from the PsychENCODE project [31]. We assessed predictive performance for both MOSTWAS and BGW-TWAS by computing McNemar's adjusted $R^2$ between observed and imputed expression.

Using ROS/MAP models, we first conducted TWAS burden testing in GWAS summary statistics for late-onset Alzheimer's disease risk from IGAP [41–44]. We subjected TWAS-identified loci at $P<2.5\times10^{-6}$ to permutation testing, and any loci that persisted past permutation testing to distal variation added-last testing. Our significance threshold is a strict Bonferroni-correction across 20,000 genes, representing the approximate protein-coding transcriptome. We compared Alzheimer's risk TWAS results from MOSTWAS with those identified by BGW-TWAS. We similarly conducted TWAS for risk of major depressive disorder (MDD) using GWAS summary statistics from PGC (excluding data from 23andMe and the UK Biobank) with the necessary follow-up tests. For any TWAS-identified loci that persisted permutation in PGC, we further conducted TWAS in GWAX summary statistics for MDD risk in the UK Biobank [46] for replication.

Using TCGA-BRCA models, we conducted TWAS burden testing [4,29] in iCOGs GWAS summary statistics for breast cancer-specific survival in a cohort of women of European ancestry. We subjected TWAS-identified loci at $P<2.5\times10^{-6}$ to permutation testing, and any locus that persisted past permutation testing to distal variation added-last testing.

## Model training and TWAS using BGW-TWAS

For comparison with MOSTWAS models trained with ROS/MAP data, we downloaded BGW-TWAS models using ROS/MAP provided by Luningham *et al* [31]. To impute expression, as described by Luningham *et al*, we multiplied the genotype matrix by BGW-TWAS weights, defined as the product of the weights and the posterior causal probabilities for each SNP.

We independently trained BGW-TWAS models in TCGA-BRCA using the same sample of 563 individuals. We first defined independent genotype blocks using LDetect [100], generated eQTL summary statistics for each block, and pruned the genome segments by selecting all local blocks and ranking distal blocks by the minimum distal-eQTL P-value in each block. We selected a maximum of 100 distal blocks with minimum distal-eQTL P-value less than 0.001. Lastly, we trained the BGW-TWAS expression model using default parameters: 3 Expectation-Maximization iterations, 10,000 burn-in MCMC iterations, 10,000 MCMC iterations, posterior causal probability threshold of 0.0001, and non-informative hyperparameters for prior causal probability ($\pi = 10^{-5}$ for both local- and distal-eQTLs) and effect size variance ($\sigma^2 = 1$ for both local- and distal-eQTLs). In TCGA, we also split the data into 3 training-test folds to calculate a cross-validation $R^2$, defined as the adjusted $R^2$ between observed and predicted values. These settings and this procedure were also employed in training models in simulation analyses.

## Supporting information

**S1 Text. Supplemental methods.** Supplemental information about mathematical and technical details of methods.
(PDF)

**S1 Fig. Comparison of TWAS power via simulations using MOSTWAS, BGW-TWAS, and local-only models.** (A) Proportion of gene-trait associations at $P<2.5\times10^{-6}$ using local-only (red), BGW-TWAS (blue), and the most predictive MOSTWAS (green) models across various local and distal expression heritability, trait heritability, and two setting of causal eQTL proportion. (B) Proportion of significant gene-trait associations across the same simulation parameters with no distal effect on the trait in the simulated external GWAS panel.
(PDF)

**S2 Fig. Comparison of predictive R2 in simulations.** Mean adjusted $R^2$ across various local and distal expression heritability, trait heritability, and causal proportions using local-only (red) and the best MOSTWAS (blue) models. The error bars reflect a width of 1 standard deviation of the 1,000 simulated adjusted $R^2$ values.
(PDF)

**S3 Fig. Comparison of false positive rates in simulations.** Boxplots of false positive rate (Y-axis) across various distal expression heritability settings (X-axis), stratified by local expression heritability (horizontal) and causal eQTL proportion (vertically) and colored by the method used to test gene-trait associations. These boxplots are from 20 simulations of 1,000 permuted phenotype traits in the simulated GWAS. The red line provides a reference at a false positive rate of 0.05.
(PDF)

**S4 Fig. Simulation analysis for the power of the distal variants added-last test.** Across various sample sizes for the eQTL reference (X-axis) panel and GWAS imputation panel (color), the power of the distal added-last test to detect a significant association with distal variants conditional on a significant local association at FDR-adjusted $P<0.05$.
(PDF)

**S5 Fig. Manhattan plot for Alzheimer's risk associations using MOSTWAS on IGAP summary statistics.** $Z$-statistic of TWAS association on the Y-axis and chromosomal position of gene on X-axis. Genes are colored red if overall $P<2.5\times10^{-6}$ and nominal permutation $P<0.05$ and labelled if distal association is significant at a Bonferroni threshold ($\alpha = \frac{0.05}{18} = 0.0028$).
(PDF)

**S6 Fig. Gene-trait associations in iCOGs and PGCs using local-only and MOSTWAS models.** $-\log_{10} P$-values of weighted burden gene-trait associations using PGC MDD risk GWAS in predominantly European-ancestry patients (A) and iCOGs survival GWAS in European-ancestry women (B) and among genes that were predicted at cross-validation $R^2 \geq 0.01$ using both local-only and MOSTWAS models and have enough SNPs in summary statistics to conduct TWAS via weighted burden test. The X- and Y-axes display the $-\log_{10} P$-values for local-only and the best MOSTWAS model, respectively. Points are colored black if P-value of association is less than or equal using the MOSTWAS model. The horizontal and vertical reference lines indicate overall Bonferroni-corrected significance thresholds ($P<2.5\times10^{-6}$).
(PDF)

**S7 Fig. Comparison of QQ-plots from TWAS associations.** QQ-plots of Z scores from TWAS for MDD in PGC (A) and breast cancer-specific survival in iCOGs (B) with local-only models (left) and MOSTWAS (right)
(PDF)

**S8 Fig. Comparison of QQ-plots from TWAS associations.** QQ-plots of $-\log_{10} P$-values from TWAS for breast cancer-specific survival in iCOGs (left) and MDD in PGC (right) with local-only models and MOSTWAS models.
(PDF)

**S9 Fig. Comparison of cross-validation $R^2$ of TCGA models using MOSTWAS and BGW-TWAS.** Scatterplot of cross-validation adjusted $R^2$ of genes using MOSTWAS (X-axis) and BGW-TWAS (Y-axis) models across 563 samples from TCGA-BRCA. The vertical and horizontal dotted lines provide reference for $R^2 = 0.01$ and the diagonal line is the 45-degree line. Each point is colored blue if the gene has a higher R2 using the MOSTWAS model, gold if

the gene has a higher $R^2$ with BGW-TWAS, and grey if the $R^2$ are equal. The proportion of models by method with $R^2 \geq 0.01$ across all imputed genes is provided.
(PDF)

**S10 Fig. Comparison of eQTL architectures across *LQK1*, *C6orf94*, and *TSC22D4*.** Manhattan plot of genome-wide eQTL associations by $-\log_{10} P$ -values (Y-axis) and chromosomal position (X-axis) at $P<10^{-3}$. SNPs included in each predictive model are highlighted in green. CV $R^2$ under each type of MOSTWAS model is provided.
(PDF)

**S11 Fig. Comparison of computation times between local-only and MOSTWAS modelling.** Mean and standard deviation of per-gene computation time across 50 randomly selected genes in TCGA-BRCA. Computations here were done in serial on a 3.0 GHz processor with 8 gigabytes of RAM.
(PDF)

**S12 Fig. Comparison of predictive ability of MeTWAS with one- and two-step regression.** Comparison of predictive performance (Y-axis) of two-step regression (labelled as MeTWAS) and one-step regression (labelled no selection) across 100 simulations across various causal proportions of eQTLs (vertically arranged), local expression heritability (horizontal), and distal expression heritability (X-axis).
(PDF)

**S13 Fig. Comparison of test power and computational speed of Sobel asymptotic and permutation tests of total mediation effect.** Power (Y-axis, left) and computational time (Y-axis, right) to detect a true large absolute total mediation effect and computation speed over various eQTL panel sample sizes (X-axis) in 10,000 simulations of mediation testing triplets.
(PDF)

**S1 Table. Comparison of $h^2$ across local-only, MeTWAS, and DePMA predictive models.** The mean and standard deviation of $h^2$ across all genes that are significantly heritable with the genetic loci considered in the design matrix of each predictive model.
(PDF)

**S2 Table. Summary statistics for known Alzheimer's risk-associated loci identified by MOSTWAS models.** TWAS associations (weighted $Z$-score and FDR-adjusted P-value) with late-onset Alzheimer's risk from GWAS statistics from IGAP. The top IGAP GWAS SNP in the identified loci with its location and P-value are provided. For the 6 loci with significant TWAS associations, the FDR-adjusted P-value for the follow-up distal SNP added last test is provided.
(PDF)

**S3 Table. Summary statistics for 18 Alzheimer's risk-associated loci identified by MOSTWAS models.** TWAS associations with Alzheimer's risk from GWAS statistics from IGAP with $P<2.5\times10^{-6}$ and permutation $P<0.05$. The top IGAP GWAS SNP in the identified loci with its location and P-value are provided.
(PDF)

**S4 Table. Summary statistics for 11 MDD risk-associated loci identified by MOSTWAS models.** TWAS associations with major depressive disorder from GWAS statistics from Psychiatric Genomics Consortium that were replicated with GWAX summary statistics in UK Biobank with permutation test results and added-last $Z$-statistics with $P<2.5\times10^{-6}$ and permutation $P<0.05$. The top PGC GWAS SNP in the identified loci with its location and P-value are

provided.
(PDF)

**S5 Table. Summary statistics for 6 breast cancer-specific survival-associated loci identified by MOSTWAS models.** TWAS associations with breast cancer survival from GWAS statistics from iCOGs with permutation test results and added-last $Z$-statistics with $P<2.5\times10^{-6}$ and permutation $P<0.05$. The top iCOGs GWAS SNP in the identified loci with its location and P-value are provided.
(PDF)

## Acknowledgments

We thank Melissa Troester, Colin Begg, Terry Furey, Michael Gandal, Sasha Gusev, Karen Mohlke, Brandon Pierce, Bogdan Pasaniuc, Hudson Santos, Jason Stein, and Cindy Wen for engaging conversation and guidance during the research process.

We thank the International Genomics of Alzheimer's Project (IGAP) for providing summary results data for these analyses. The investigators within IGAP contributed to the design and implementation of IGAP and/or provided data but did not participate in analysis or writing of this report. IGAP was made possible by the generous participation of the control subjects, the patients, and their families. The iSelect chips were funded by the French National Foundation on Alzheimer's disease and related disorders. EADI was supported by the LABEX (laboratory of excellence program investment for the future) DISTALZ grant, Inserm, Institut Pasteur de Lille, Université de Lille 2 and the Lille University Hospital. GERAD was supported by the Medical Research Council (Grant n˚ 503480), Alzheimer's Research UK (Grant n˚ 503176), the Wellcome Trust (Grant n˚ 082604/2/07/Z) and German Federal Ministry of Education and Research (BMBF): Competence Network Dementia (CND) grant n˚ 01GI0102, 01GI0711, 01GI0420. CHARGE was partly supported by the NIH/NIA grant R01 AG033193 and the NIA AG081220 and AGES contract N01–AG–12100, the NHLBI grant R01 HL105756, the Icelandic Heart Association, and the Erasmus Medical Center and Erasmus University. ADGC was supported by the NIH/NIA grants: U01AG032984, U24 AG021886, U01 AG016976, and the Alzheimer's Association grant ADGC–10–196728.

We also thank the Psychiatric Genomics Consortium for their publicly available GWAS summary statistics (https://www.med.unc.edu/pgc/) and the Pickrell Lab at the New York Genome Center for their GWAS browser and GWAX summary statistics (http://gwas-browser.nygenome.org/downloads/gwas-browser/). We thank the PsychENCODE project for sharing their eQTL dataset for external validation of models. Data were generated as part of the PsychENCODE Consortium supported by: U01MH103339, U01MH103365, U01MH103392, U01MH103340, U01MH103346, R01MH105472, R01MH094714, R01MH105898, R21MH102791, R21MH105881, R21MH103877, and P50MH106934 awarded to: Schahram Akbarian (Icahn School of Medicine at Mount Sinai), Gregory Crawford (Duke), Stella Dracheva (Icahn School of Medicine at Mount Sinai), Peggy Farnham (USC), Mark Gerstein (Yale), Daniel Geschwind (UCLA), Thomas M. Hyde (LIBD), Andrew Jaffe (LIBD), James A. Knowles (USC), Chunyu Liu (UIC), Dalila Pinto (Icahn School of Medicine at Mount Sinai), Nenad Sestan (Yale), Pamela Sklar (Icahn School of Medicine at Mount Sinai), Matthew State (UCSF), Patrick Sullivan (UNC), Flora Vaccarino (Yale), Sherman Weissman (Yale), Kevin White (UChicago) and Peter Zandi (JHU).

We thank the members of the ROSMAP project for providing their publicly available data. The ROSMAP project was supported by funding from the National Institute on Aging (AG034504 and AG041232).

Funding for BCAC and iCOGS came from: Cancer Research UK [grant numbers C1287/A16563, C1287/A10118, C1287/A10710, C12292/A11174, C1281/A12014, C5047/A8384, C5047/A15007, C5047/A10692, C8197/A16565], the European Union's Horizon 2020 Research and Innovation Programme (grant numbers 634935 and 633784 for BRIDGES and B-CAST respectively), the European Community's Seventh Framework Programme under grant agreement n˚ 223175 [HEALTHF2-2009-223175] (COGS), the National Institutes of Health [CA128978] and Post-Cancer GWAS initiative [1U19CA148537, 1U19 CA148065-01 (DRIVE) and 1U19 CA148112—the GAME-ON initiative], the Department of Defense [W81XWH-10-1-0341], and the Canadian Institutes of Health Research CIHR) for the CIHR Team in Familial Risks of Breast Cancer [grant PSR-SIIRI-701]. All studies and funders as listed in Michailidou K *et al* (2013 and 2015) and in Guo Q et al (2015) are acknowledged for their contributions.

## Author Contributions

**Conceptualization:** Arjun Bhattacharya, Michael I. Love.

**Data curation:** Arjun Bhattacharya.

**Formal analysis:** Arjun Bhattacharya.

**Funding acquisition:** Yun Li, Michael I. Love.

**Methodology:** Arjun Bhattacharya, Michael I. Love.

**Software:** Arjun Bhattacharya.

**Supervision:** Michael I. Love.

**Validation:** Arjun Bhattacharya, Yun Li, Michael I. Love.

**Visualization:** Arjun Bhattacharya, Yun Li, Michael I. Love.

**Writing – original draft:** Arjun Bhattacharya.

**Writing – review & editing:** Arjun Bhattacharya, Yun Li, Michael I. Love.

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
