## [Decision Letter · Decision Letter 0]

25 Oct 2020

Dear Dr Love,

Thank you very much for submitting your Research Article entitled 'MOSTWAS: Multi-Omic Strategies for Transcriptome-Wide Association Studies' to PLOS Genetics. Your manuscript was fully evaluated at the editorial level and by independent peer reviewers and the editors. Based on the reviews, we will not be able to accept this version of the manuscript, but we would be willing to review again a much-revised version. We cannot, of course, promise publication at that time.

Although both reviewers found the work to be interesting, they  raised substantial concerns. Both reviewers noted that TWAS approaches using cis- and trans-eQTLs have been published. Therefore, further comparisons of the proposed methods (MeTWAS amd DePMA) with existing methods should be performed to demonstrate the advances of the proposed approaches in both simulations and real data analysis, rather than only a broad discussion.  In addition, a definition of distal-eQTL with a distance of 0.5Mb or more is not consistent with the literature.  We would also like to see the effect of using the more standard definition.

If you decide to revise the manuscript for further consideration at PLOS Genetics, please aim to resubmit within the next 60 days, unless it will take extra time to address the concerns of the reviewers, in which case we would appreciate an expected resubmission date by email to plosgenetics@plos.org.

[LINK]

We are sorry that we cannot be more positive about your manuscript at this stage. Please do not hesitate to contact us if you have any concerns or questions.

Yours sincerely,

Xiaofeng Zhu

Associate Editor

PLOS Genetics

David Balding

Section Editor: Methods

PLOS Genetics

Reviewer's Responses to Questions

**Comments to the Authors:**

Reviewer #1: Bhattacharya et al. developed two novel TWAS (Transcriptome-Wide Association Studies) methods to incorporate multi-omics data: mediator-enriched TWAS (MeTWAS) and distal-eQTL prioritization via mediation analysis (DePMA). As compared to most existing TWAS methods that only relies on local-eQTLs, the two proposed methods borrow additional information from distal-eQTLs and other omics data such as DNA methylation and microRNA. Both methods adopt rigorous mediation analysis approaches. Below I list some minor issues that may further improve the manuscript.

1. Incorporating omics data in TWAS has become a hot research topic recently. For instance, the following related paper, initially been posted on bioRxiv on March 6, also considers distal-eQTLs in TWAS.

“Bayesian Genome-wide TWAS method to leverage both cis- and trans-eQTL information through summary statistics”

https://www.biorxiv.org/content/10.1101/2020.03.05.979187v2

It is worth discussion if a comparison is too much work. It is especially key for reproducibility across methods. A quick look does not find overlap genes for Alzheimer’s disease.

2. Which local-only methods were used for comparison in simulations and data analysis? Given different local-only algorithms, it would be precise to explicitly note what exact method was used each time when local-only models were mentioned.

3. The authors defined distal-eQTL with a distance of 0.5 Mb, while many other studies used a distance of 1 Mb. I am wondering if this would affect the performance of the proposed TWAS methods and biological interpretation.

4. Some figures need improvement.

Fig 3, making each subfigure square would facilitate reading;

Fig 4 AB, no legends for the three colors. What do P and D mean in Fig 4B?

5. There are some typos.

Line 70, “many distal-eQTLs are often eQTLs for one of more of their local genes”

“one of more of” should be “one or more of”?

In the section of “real data applications in brain tissue,” “tumor” appeared multiple times. It should be avoided because the data are about Alzheimer’s disease.

Line 400, typo “sties”.

Line 472, redundant words, “we considered two scenarios.”

Supplemental Methods, formula (2), the second equation, \\beta_M should be w_G?

Reviewer #2: The authors proposed two approaches to consider potential distal-eQTL for improving the prediction accuracy of gene expression and then the TWAS power. Here are my comments:

1. Although it is good to see that additional distal-eQTL could play an important role in TWAS, the authors did not describe their methods clearly in the main text. I think it would be helpful to include brief formulas to introduce the models of these two proposed methods in the main text. Include Figure S1 as part of Figure 1 would also help.

2. The authors shall also compare with the recently published BGW-TWAS method (https://doi.org/10.1016/j.ajhg.2020.08.022) that considers both cis- and trans- eQTL for TWAS, and used the same ROS/MAP application data set.

3. It is not clearly stated about the number of mediators of different omics- types that were considered in the application studies. For the gene expression trait of each gene, will all mediators available for the study cohort be considered by MeTWAS and DePMA?

4. For the computation time mentioned in the main text, it is not clear whether additional computation time about generating standard eQTL p-values, or identifying associated mediators are included or not.

5. The authors need to be specific about which local-only models were compared in the application studies, as PrediXcan, Fusion, and TIGAR use different statistical models and perform differently.

6. In the application study of using ROS/MAP data, as mentioned from lines 199-201, does it mean only 267 genes has training R2 > 0.01 by local-only models, 911 by MeTWAS, and 2934 by DePMA? The number of genes is much less than reported in the TIGAR paper.

7. Similar number of genes with training R2>0.01 were also identified for the transcriptomic profiles of breast tissue type. This is also much less than what was published for studying breast cancer using TCGA data (https://www.nature.com/articles/s41588-018-0132-x). The number of significant TWAS genes were also much less than published for using the same BRCA GWAS data set.

8. A discussion about which TWAS method (MeTWAS or DePMA) would be preferred by what type of genetic architecture, or multi-omics network for the test gene will be helpful.

9. Color legend need to be included for Figure 4A, 4B.

**Have all data underlying the figures and results presented in the manuscript been provided?**

Reviewer #1: Yes

Reviewer #2: Yes

PLOS authors have the option to publish the peer review history of their article (what does this mean?). If published, this will include your full peer review and any attached files.

Reviewer #1: No

Reviewer #2: No

---

## [Decision Letter · Decision Letter 1]

20 Jan 2021

Dear Dr Love,

Thank you very much for submitting your Research Article entitled 'MOSTWAS: Multi-Omic Strategies for Transcriptome-Wide Association Studies' to PLOS Genetics.

The manuscript was fully evaluated at the editorial level and by independent peer reviewers. Reviewer 1 has been satisfied but reviewer 2 identified some minor concerns that we ask you address in a revised manuscript

We therefore ask you to modify the manuscript according to the review 2's recommendations. Your revisions should address the specific points made by the reviewer.  In particular we encourage you to put an outline of the method before the results, this deviates from the traditional PLOS Genetics format but is now encouraged for methods papers.

[LINK]

Yours sincerely,

Xiaofeng Zhu

Associate Editor

PLOS Genetics

David Balding

Section Editor: Methods

PLOS Genetics

Reviewer's Responses to Questions

**Comments to the Authors:**

Reviewer #1: My comments have been addressed.

Reviewer #2: The authors have addressed most of my comments. I only have the following minor comments:

1. For the R2 comparison between BGW-TWAS and MOSTWAS, the figure showed there are more number of genes have greater R2 by BGW-TWAS method. It is not fair to compare the proportion of genes that have greater R2 as the total number of gene models fitted by BGW-TWAS method is >twice of the ones by MOSTWAS.

2. Please note that the BGW-TWAS computation time include single-variant eQTL test for genome-wide genotypes.

3. I think MOSTWAS might have an advantage over BGW-TWAS by extensively testing the mediation effect for some genes, but might miss some mediating effects that are too weak to be detected. The number of gene models fitted by MOSTWAS now is approximately similar to the so called local method but smaller than the BGW-TWAS method, which is mainly due to the Elastic-Net regression model. This could be improved if the Bayesian DPR model as used by TIGAR is used to fit the gene expression prediction model.

4. Since MOSTWAS may not be able to screen all other genome-wide multi-omics factors for each test gene, it would help to discuss how to select a set of “mediating omics factors” to start. And make it clear that if users need to first conduct single variant eQTL analysis for genome-wide genotypes per gene to identify all potential distal eQTL for mediation test?

5. I feel like a simplified method section could go ahead of the Results section to help reader to understand the method and results better. And more details could go to supplement. The editors can see if this would make sense.

**Have all data underlying the figures and results presented in the manuscript been provided?**

Reviewer #1: None

Reviewer #2: Yes

PLOS authors have the option to publish the peer review history of their article (what does this mean?). If published, this will include your full peer review and any attached files.

Reviewer #1: No

Reviewer #2: No

---

## [Decision Letter · Decision Letter 2]

4 Feb 2021

Dear Dr Love,

We are pleased to inform you that your manuscript entitled "MOSTWAS: Multi-Omic Strategies for Transcriptome-Wide Association Studies" has been editorially accepted for publication in PLOS Genetics. Congratulations!

Yours sincerely,

Xiaofeng Zhu

Associate Editor

PLOS Genetics

David Balding

Section Editor: Methods

PLOS Genetics

Comments from the reviewers (if applicable):

Reviewer's Responses to Questions

**Comments to the Authors:**

Reviewer #2: All of my comments are addressed.

**Have all data underlying the figures and results presented in the manuscript been provided?**

Reviewer #2: None

PLOS authors have the option to publish the peer review history of their article (what does this mean?). If published, this will include your full peer review and any attached files.

Reviewer #2: No

**Data Deposition**

http://datadryad.org/submit?journalID=pgenetics&manu=PGENETICS-D-20-01017R2

**Press Queries**

---

## [Editor Report · Acceptance letter]

23 Feb 2021

PGENETICS-D-20-01017R2 

MOSTWAS: Multi-Omic Strategies for Transcriptome-Wide Association Studies 

Dear Dr Love, 

We are pleased to inform you that your manuscript entitled "MOSTWAS: Multi-Omic Strategies for Transcriptome-Wide Association Studies" has been formally accepted for publication in PLOS Genetics! Your manuscript is now with our production department and you will be notified of the publication date in due course.

With kind regards,

Alice Ellingham

PLOS Genetics

On behalf of:
